# Do Urban Food Deserts Exist in the Global South? An Analysis of Nairobi and Mexico City

**Jeremy Wagner** [1],*, **Lucy Hinton** [1], **Cameron McCordic** [2], **Samuel Owuor** [3], **Guénola Capron** [4] **and Salomón Gonzalez Arellano** [5]

1   Balsillie School of International Affairs, 67 Erb Street West, Waterloo, Ontario, N2L 6C2, Canada; lhinton@balsillieschool.ca

2   School of Environment, Enterprise and Development (SEED), Faculty of Environment, University of Waterloo, 200 University Avenue West, Waterloo, Ontario, N2L 3G1, Canada; c2mccordic@uwaterloo.ca

3   Department of Geography & Environmental Studies, University of Nairobi, Hyslop Building, Main Campus, Nairobi, P.O.Box 30197-00100, Kenya; samowuor@uonbi.ac.ke

4   División de Ciencias Sociales, Departamento de Sociología, Universidad Autónoma Metropolitana – Azcapotzalco, Av San Pablo Xalpa 180, Reynosa Tamaulipas, 02200 Ciudad de México, CDMX, Mexico; guenola.capron@gmail.com

5   Planta Académica de Ciencias Sociales, Universidad Autónoma Metropolitana – Cuajimalpa, Vasco de Quiroga 4871, Contadero, 05370 Cuidad de México, CDMX, Mexico; salomonglez@gmail.com

*   Correspondence: jwagner@balsillieschool.ca

**Abstract:** Recent conceptualizations of 'food deserts' have expanded from a sole focus on access to supermarkets, to food retail outlets, to all household food sources. Each iteration of the urban food desert concept has associated this kind of food sourcing behavior to poverty, food insecurity, and dietary diversity characteristics. While the term continues to evolve, there has been little empirical evidence to test whether these assumed associations hold in cities of the Global South. This paper empirically tests the premises of three iterations of the urban food desert concept using household survey data collected in Nairobi, Kenya, and Mexico City, Mexico. While these associations are statistically significant and show the expected correlation direction between household food sourcing behavior and food security, the strength of these relationships tends to be weak. These findings indicate that the urban food desert concept developed in North American and UK cities may have limited relevance to measuring urban food insecurity in the Global South.

**Keywords:** food deserts; food security; food sourcing; supermarkets; dietary diversity; Mexico City; Nairobi

## 1. Introduction

Since the 1990s, the 'food desert' concept has been extensively used in cities in the Global North, most often in the United States, Canada, and the United Kingdom. It has not, however, been applied systematically to cities of the Global South. While there are many potential reasons for this gap, chief among them is the way food security and malnutrition in the Global South are often framed as rural issues and related to hunger and food scarcity [1]. This particular food security paradigm is present in both research and policy discourse and has limited discussions on urban food security in the South until recently. In addition, food deserts have traditionally been related to the presence or absence of supermarkets, which, while proliferating, do not yet have a commanding presence in urban food systems in many parts of the Global South [2,3]. Researchers who use the food desert concept argue that differences in food access between households and neighborhoods can be best understood

through the structural and spatial dimensions of food environments. To date, analyses of the food desert concept have not been systematically applied to cities in the Global South.

Food deserts are usually characterized as economically disadvantaged areas where there is relatively poor access to healthy and affordable food because of the absence of modern retail outlets [4]. Cities in the Global South contain many poor neighborhoods where the prevalence of malnourishment and food insecurity are often far more dramatic than in North America and the UK. Therefore, the key question is whether mainstream definitions of food deserts applied to the Global North can be usefully applied to the Global South's rapidly growing cities. If the concept can be reformulated to fit the realities of urban food systems in the Global South, 'food deserts' may prove to be a useful analytical tool which urban food researchers and policy makers have yet to fully explore.

In order to perform a preliminary test of the applicability of the food desert concept in the Global South, this paper uses household survey data collected in Nairobi and Mexico City by the Hungry Cities Partnership. The paper first provides an account of three iterations the food desert concept that that are based on the authors' interpretations of the literature.

### 1.1. Classic Food Deserts

The concept of a food desert grew out of a small but growing body of evidence that suggested food items may be more difficult to access in deprived areas [5–7]. While policy interventions were undertaken by the UK government, there was a dearth of evidence on the causal factors of food deserts. Originally, conceptions of food deserts were based primarily on distance to supermarkets. The further a neighborhood was from a supermarket, the larger the food desert was considered to be. An absence of supermarkets in a neighborhood was deemed a result of redlining: a spatially discriminatory practice among retailers of not serving certain areas, based on their demographic composition [8]. These same neighborhoods were sometimes characterized as 'too low-income', leaving retailers concerned with profitability. This first version of the food desert was tied to highly quantitative, easily calculable values like distances and food prices. However, without significant evidence to endorse the quantifiable variables being used, the food desert concept has since evolved in its application to cities in the Global North.

### 1.2. Food Deserts Plus

More recent conceptualizations of food deserts are characterized by recognition and acceptance of the nuanced nature of food accessibility in a city. In this iteration, food deserts are no longer considered simply a spatial issue, to be analyzed through the addition of more variables. Instead, the food desert is seen as a dynamic meshwork of social, economic, and political interactions [9–11]. Studies increasingly consider the interrelated nature of income, mobility, transportation, time, seasonality, family structure, presence of different type of retail location, dietary diversity, education, structural inequalities, and so on [12,13]. Policy environments that shape inequalities in neighborhood access to food have also been further explored. Perhaps the most important shift was the growing understanding that distance to supermarkets was a proxy measure for food access, and that this may be an inadequate way to measure how marginalized populations were actually eating.

In this iteration of the food desert concept, the inclusion of dietary diaries into research methodologies was popularized, and many studies underlined the need to grasp the 'healthiness' of foods being accessed [13]. This gave rise to concepts such as the 'food oasis' (pockets of healthy food access) and the 'food swamp' (an abundance of unhealthy food), further elaborating on the spatial component of inequality in food access in urban areas. Food deserts became more complex conceptualizations, and fruit and vegetable consumption became nearly as ubiquitous as supermarket analysis had once been. Food Deserts Plus represents a more recent understanding of the food desert concept reflected within the literature and remains predominantly applied to cities in the Global North.

*1.3. Food Deserts in the Global South*

Along with more complex and nuanced understandings of food deserts emerging, researchers have also began providing supplemental perspectives on food deserts by using empirical evidence to test whether these assumed associations hold in cities of the Global South. There are, of course, reasons for caution when applying a Euro-American understanding of food deserts in cities with histories and geographies of urban food retailing and food system development that are remarkably different. For one, the importance of the informal food economy for residents' food security in growing cities in the South poses a set of challenges to conventional approaches measuring food deserts. Informal food vending is fluid and dynamic, and retailers might relocate their business frequently. Retail typologies and geographies are therefore significantly different, and lack of access to a supermarket is potentially less important of a factor in facilitating neighborhood food insecurity. Supermarkets are, of course, an increasingly important retail type in many Southern cities globally, but a sole focus on modern retailing cannot capture all of the market and nonmarket food sources, or the spatial mobility of informal retailing. With important contextual differences to be taken into account, Crush and Battersby redefine food deserts in the Global South as "poor, often informal, urban neighborhoods characterized by high food insecurity and low dietary diversity, with multiple market and nonmarket food sources but variable household access to food" [4]. As such, this adapted concept has potential to be a useful analytical tool to understand the structural barriers leading to inequalities in food access in the Global South.

This paper intends to add to the growing body of literature on urban food deserts in the Global South by empirically assessing these three conceptualizations of urban food deserts in the contexts of Nairobi and Mexico City. The following sections evaluate the usefulness of three definitions when applied to these contexts. By examining two cities from different geographical areas, this paper explores how differing urban food environments in the Global South affect the relevance and findings of the three food desert iterations tested. Following an analysis and discussion of the results, the paper concludes by highlighting the research and policy implications.

## 2. Materials and Methods

Mexico City and Nairobi were initially selected as study sites because of their distinct contexts as cities located in the Global South—a geographic area where the concept of food deserts has not been systematically applied. Each city and their food systems have contrasting cultural, structural, and development patterns. These location selections provide an opportunity to operationalize food desert definitions and utility in unique settings and provide comparative analyses between the two.

Nairobi is a relatively young city experiencing a rapid rate of urbanization that is stretching existing food and agriculture systems, now struggling to provide food and nutrition security for inhabitants [14]. In 2009, the population of Nairobi was 3.1 million, with projections estimating this number to double by 2025 [15]. As a result, the city is dynamic and growing, and its food supply chains are always adapting to changing local conditions. Kenya's domestic food supply chain system is a significant contributor to the economy: the agricultural sector is 26% of national GDP [16]. Informal traditional value chains continue to play a vital role in food provisioning throughout the city. These chains are characterized by the variety of actors and intermediaries that increase transaction costs, creating an inefficient postharvest procurement network, and thereby pushing food products out of reach for those who need them most. Local authorities have used by-laws and regulations to suppress the development of street vending and other forms of informal trade. As a result, Kenya's informal economy has often been subject to policies that produce unfavourable business environments. Even so, the informal food economy is dynamic and persists as a central source of food for the cities inhabitants [14].

Nairobi's formalized food system is expanding as well and relies on centralized and regionalized procurement networks, specialized wholesalers and supplier systems, and modern retailing outlets that seek competitive advantage through direct control of their procurement systems [14]. This trend

towards a formalized food system development is recent however, and the impact it will have on food access, neighborhood food environments, and the city's vitally important food markets and associated informal sector within the city remains unclear.

In contrast, Mexico City is a larger city with a much longer history. Meeting the daily food demands of Mexico City's over 20 million inhabitants requires food products to be procured from a combination of traditional and highly sophisticated modern food supply chain systems from rural areas, its fishing industry, and food imports. While traditional food systems are still vital for food provisioning in the city, Capron et al. suggest that the diets of residents of Mexico's capital city are increasingly influenced by food economies of developed countries [17]. As of 2016, supermarkets, grocers, and corner stores control 52% of food sales in the Mexico City Metropolitan Area. Diets are characterized as nutrient-poor, energy-dense, and highly processed, and these characteristics are associated with growing obesity, overnutrition, and micronutrient deficiencies. Between 2000 and 2012, adult obesity increased from 16% to 26% of the city's inhabitants [17].

The geography of informal food markets and vending in Mexico City is increasingly subject to restrictive state policies related to public space, streets, plazas and parks, as well as permissive state-led, market-oriented policies towards the redevelopment of specific commercial, residential and tourist zones in downtown Mexico City through up-zoning, and changes to land use regulations and area-specific plans [17]. These policies have combined to restrict or displace the informal vending activities, particularly in high-income and central areas of the city.

In this sense, Nairobi and Mexico City as study sites reflect some of the similarities but also significant differences observed between urban food systems in the Global South. These location selections provide an opportunity to operationalize food desert conceptions in contrasting urban settings and comparatively analyze the two sets of findings.

## 2.1. Research Objectives and Questions

Research Objectives and Questions.

| Research Objectives | Research Questions |
|---|---|
| Objective 1: Test the Original Concept of Food Deserts | 1.1 Is there a relationship between household supermarket access and household food security? |
| | 1.2 Is there a relationship between household poverty and supermarket access? |
| Objective 2: Test Emerging Concepts of Food Deserts (Food Deserts Plus) | 2.1 Is there a relationship between household access to all food retail sources and household food security? |
| | 2.2 Is there a link between the type food products purchased and the sources of those food products at the household level? |
| | 2.3 Is there a link between fruit and vegetable purchase/consumption and household food security? |
| | 2.4 Is the number of household food retail sources related to household food security? |
| Objective 3: Test Crush and Battersby's (2017) definition of food deserts (Food Deserts in the Global South) | 3.1 Is there a relationship between access to all food sources (market and otherwise) and household income, household dietary diversity, food access/food price challenges? |

## 2.2. Sampling

The data used to answer these questions is drawn from household survey data from Nairobi and Mexico City in 2016. In Nairobi the household sample was stratified by subdistrict population, with subdistricts randomly selected from within all districts in Nairobi City County (Figure 1). Households were then selected by enumerator teams within each subdistrict using systematic sampling, resulting in a final sample size of 1424 households.

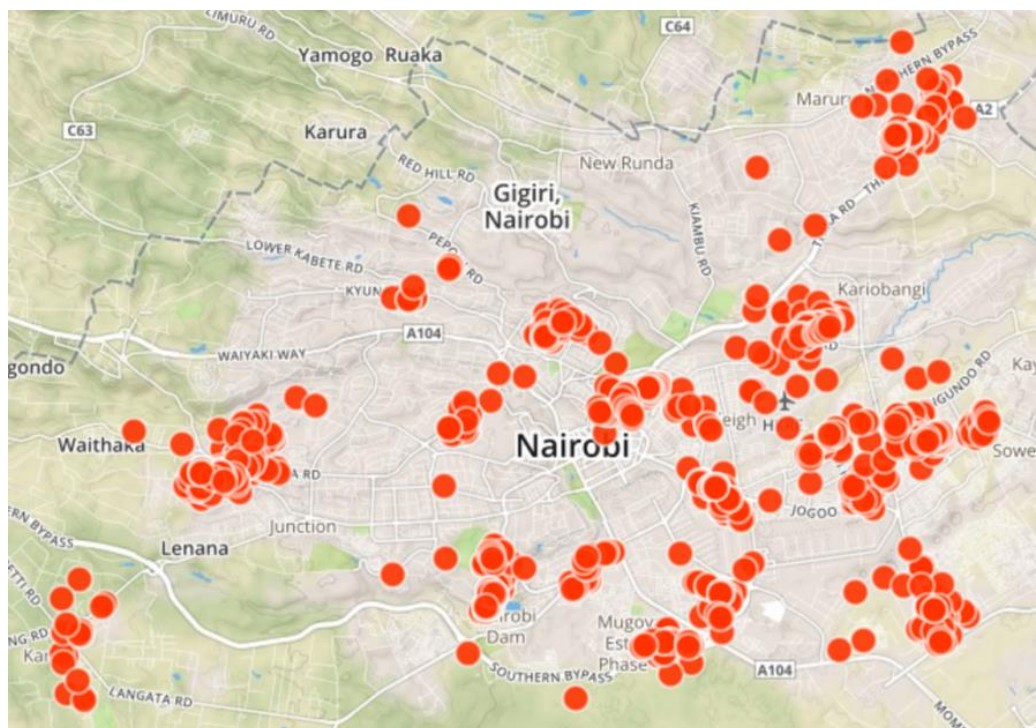

**Figure 1.** Nairobi sampling distribution.

In the household survey of Mexico City, enumeration areas were randomly selected across the entire metropolitan area (Figure 2). The total sample size was stratified using proportionate allocation across these enumeration areas within socioeconomic bands. Households were then selected by teams of enumerators using random systematic sampling. The total sample size for this survey was 1210 households.

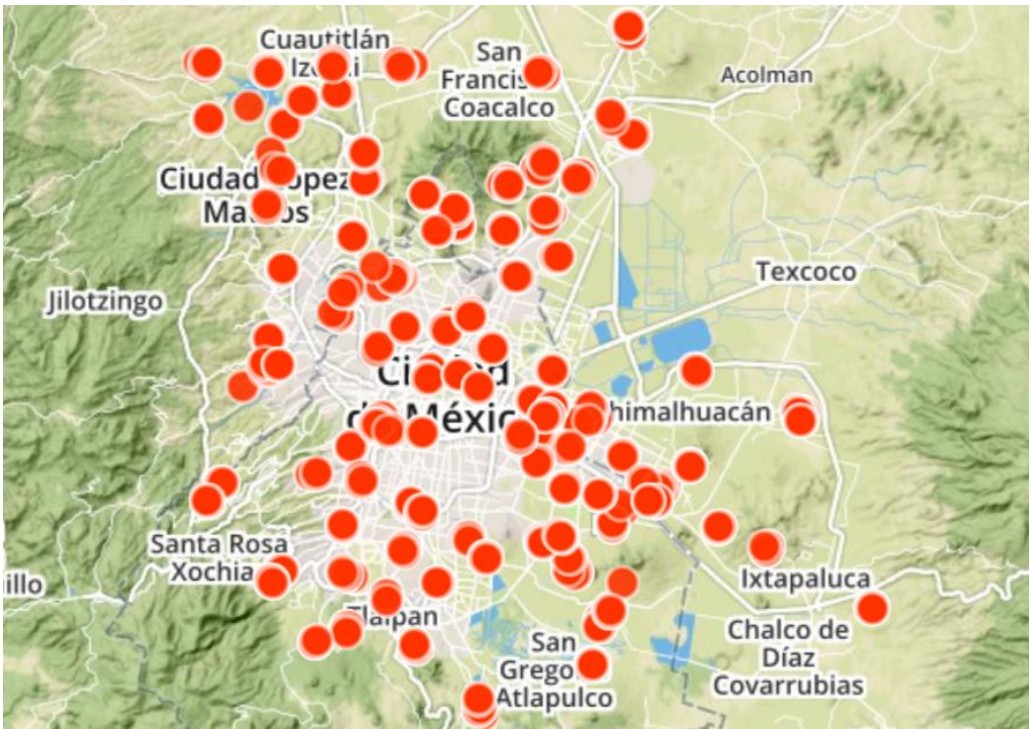

**Figure 2.** Mexico City sampling distribution.

*2.3. Measures*

These citywide surveys of Mexico City and Nairobi used the Hungry Cities Partnership (HCP) household survey instrument. This survey instrument measured household food security and food sourcing behavior, together with measures of poverty and demographic characteristics. This paper relies on the following measures taken from this survey instrument: the Household Food Insecure Access Prevalence scale (HFIAP), the Household Dietary Diversity Score (HDDS), the Lived Poverty Index (LPI), household income quintiles, food price impact, and household food sources.

The HFIAP is an ordinal-level scale that measures the severity of household food insecurity in the last month [18]. The score is calculated using nine Likert scale questions on the frequency with which households have experienced various dimensions of food access challenges in the last four weeks. The answers to these questions are then aggregated using a scoring algorithm to classify households according to four categories of food insecurity: Food Secure, Mildly Food Insecure, Moderately Food Insecure, and Severely Food Insecure.

The HDDS is an ordinal-level scale that represents the number of food groups that have been consumed by any member of the household in the last 24 hours (with a total of 12 possible food groups included in the scale) [19]. A higher score on the HDDS indicates greater dietary diversity. The LPI is an ordinal-level scale that measures lived poverty. The scale is made up of six Likert scale questions measuring the frequency with which households have gone without electricity, clean water, medical care, cooking fuel, food, or a cash income in the last year. The LPI scale score is the average of these six subscale questions. A higher score on the LPI represents greater severity of lived poverty.

Household income quintiles were calculated by summing the amount of income earned by households in the last month across all household income sources (except for any loans or credit). This total household income was then binned into five ranked and proportionately equal categories or quintiles. This calculation was done within each city and not across both cities. A higher score on the household income quintiles represents higher household income.

The HCP household survey instrument also included a question on food price impact. In this question, the respondent was asked to indicate the frequency with which his or her household went without food due to rising food prices in the previous six months. The response was recorded using a Likert scale indicating frequency of occurrence. The household food sources measured in the survey instrument indicate the source of food accessed by the household in the previous month. The food items and food sources measured in each survey varied slightly to ensure that contextually important food sources were recorded in the survey instrument.

*2.4. Analysis*

To achieve this study's research objectives, this investigation uses frequency distributions, measures of central tendency, Pearson's chi-squared test of independence, Fisher's exact test, and Spearman's Rho correlation. All analyses were carried out using the SPSS version 24 statistical package.

Some of the assessments of the relationships between the variables included in this investigation use cross-tabulations. These cross-tabulations represent both measures of central tendency (averages) across the categories of other variables as well as frequency distributions. The frequency distributions included in this investigation are assessed using Pearson's chi-squared test of independence. This test determines whether the distributed frequency of households across any two categorical variables is random. In the case where the assumptions of this test are violated, the Fisher's exact test is used to test for the association between two categorical variables. In this investigation, Pearson's chi-squared test of independence was used to assess the relationship between the HFIAP and supermarket access and the LPI and supermarket access (to achieve research objective 1). These tests were also applied to cross-tabulations of the HFIAP by household fruit and vegetable consumption (to achieve research objective 2). The mean HFIAS scores were also calculated according to different household food sources (to achieve research objective 2).

Spearman's Rho correlations determine the extent to which two ordinal or continuous-level variables are related. This correlation is sensitive to nonlinear relationships but also assumes a monotonic relationship (continuously increasing or decreasing relationships). The Spearman's Rho correlation strength is assessed in this investigation using the criteria according to Prion and Haerling [20], where <0.4 indicates a negligible or weak relationship, 0.4–0.6 indicates a moderate relationship, and >0.6 indicates a strong relationship. In this investigation, the HFIAP, HDDS, household food price impact, and household income quintiles were all correlated with the number household food sources (to achieve research objective 3).

## 3. Results

### 3.1. Objective One Results: Test the Original Concept of Food Deserts

#### 3.1.1. Is There a Relationship between Household Supermarket Access and Household Food Security?

This investigation found a statistically significant but weak relationship between household supermarket access the year prior to the survey and household food security status in Nairobi. As indicated in Table 1, these variables share a nonrandomly distributed relationship according to a chi-squared test of independence at an alpha of 0.05 ($x2 = 73.509$, $p < 0.001$, n = 1401). Among the sampled households in Nairobi that accessed supermarkets in the last year, there was a negligible but statistically significant Spearman's Rho correlation of 0.193 (n = 1093, $p < 0.001$) between frequency of supermarket access and household food security status (Table 1). Only 14.4% of those who did not access supermarkets in the last year were food secure, whereas 41.8% of those with no access were severely food insecure; 33.2% of those who accessed supermarkets in the last year were food secure, while only 20.7% with access were severely food insecure.

**Table 1.** Household Food Insecure Access Prevalence scale (HFIAP) scores and supermarket access in previous year in Nairobi.

| Food Security Status | no access n (%) | access n (%) |
|---|---|---|
| Food Secure | 43 (14.4%) | 366 (33.2%) |
| Mildly Food Insecure | 28 (9.4%) | 148 (13.4%) |
| Moderately Food Insecure | 103 (34.4%) | 360 (32.7%) |
| Severely Food Insecure | 125 (41.8%) | 228 (20.7%) |
| Total | 299 (100%) | 1102 (100%) |

There was a similarly significant but weak relationship between household supermarket access and food security in Mexico City. Table 2 indicates that these variables share a nonrandomly distributed relationship according to a chi-squared test of independence at an alpha of 0.05 ($x2 = 74.933$, $p < 0.001$, n = 1200) and a statistically significant Spearman's Rho correlation of 0.138 (n = 681, $p = 0.006$) (Table 2). A majority (59.9%) of those who accessed supermarkets in the last year were food secure while only 19.3% with access were severely food insecure. A total of 36.0% of those who did not access supermarkets in the last year were food secure, whereas 36.6% of those with no access were severely food insecure.

**Table 2.** HFIAP scores and supermarket access in previous year in Mexico City.

| Food Security Status | no access n (%) | access n (%) |
|---|---|---|
| Food Secure | 186 (36.0%) | 409 (59.9%) |
| Mildly Food Insecure | 66 (12.8%) | 80 (11.7%) |
| Moderately Food Insecure | 76 (14.7%) | 62 (9.1%) |
| Severely Food Insecure | 189 (36.6%) | 132 (19.3%) |
| Total | 517 (100%) | 683 (100%) |

These observations also extend to regular (monthly) household supermarket access. In Nairobi, regular supermarket access shared a nonrandomly distributed relationship with household food security according to a chi-squared test of independence at an alpha of 0.05 ($x2 = 132.596$, $p < 0.001$, n = 1382) (Table 3). Over a third (35.2%) of households with regular access to supermarkets were food secure, while only 17.7% of households were severely food insecure. Among the households that irregularly accessed supermarkets, 43.8% were severely food insecure, while only 13.9% were food secure.

**Table 3.** HFIAP scores and regular supermarket access in Nairobi.

| Food Security Status | regular access n (%) | irregular access n (%) |
|---|---|---|
| Food Secure | 345 (35.2%) | 56 (13.9%) |
| Mildly Food Insecure | 143 (14.6%) | 31 (7.7%) |
| Moderately Food Insecure | 319 (32.6%) | 139 (34.6%) |
| Severely Food Insecure | 173 (17.7%) | 176 (43.8%) |
| Total | 980 (100%) | 402 (100%) |

Similarly, in Mexico City, there is not a significant difference in the relationship between household food security scores and regular versus irregular supermarket access. Table 4 indicates a nonrandomly distributed relationship according to a chi-squared test of independence at an alpha of 0.05 ($x2 = 66.660$, $p < 0.001$, n = 1190). As many as 56.5% of households with regular access to supermarkets were food secure and only 20.7% were severely food insecure. Among the households that irregularly accessed supermarkets, 39.6% were severely food insecure, while 34.3% were food secure.

**Table 4.** HFIAP scores and regular supermarket access in Mexico City.

| Food Security Status | regular access n (%) | irregular access n (%) |
|---|---|---|
| Food Secure | 450 (56.5%) | 135 (34.3%) |
| Mildly Food Insecure | 101 (12.7%) | 45 (11.4%) |
| Moderately Food Insecure | 80 (10.1%) | 58 (14.7%) |
| Severely Food Insecure | 165 (20.7%) | 156 (39.6%) |
| Total | 796 (100%) | 394 (100%) |

3.1.2. Is There a Relationship between Household Poverty and Supermarket Access?

The sampled households in Nairobi indicated a significant, but weaker, relationship between supermarket access and LPI. Table 5 shows that households that accessed supermarkets shared a nonrandomly distributed relationship with the LPI according to a Fisher's exact test of independence at an alpha of 0.05 (F = 42.866, $p < 0.001$, n = 1351) and a negligible but statistically significant Spearman's Rho correlation of 0.074 (n = 1067, $p = 0.015$) with LPI. 91.1% of households who accessed a supermarket in the past year had an LPI of 1.00 or less, compared with 77.6% of households who did not access a supermarket.

**Table 5.** Lived poverty and supermarket access in previous year in Nairobi.

| Lived Poverty Status | no access n (%) | access n (%) |
|---|---|---|
| <=1.00 | 215 (77.6%) | 978 (91.1%) |
| 1.01–2.00 | 52 (18.8%) | 93 (8.7%) |
| 2.01–3.00 | 9 (3.2%) | 3 (0.3%) |
| 3.01+ | 1 (0.4%) | 0 (0.0%) |
| Total | 277 (100%) | 1074 (100%) |

The relationship was even weaker in Mexico City. As Table 6 shows, household supermarket access shared a nonrandomly distributed relationship with the LPI according to a Fisher's exact test of independence at an alpha of 0.05 (F = 24.082, *p* < 0.001, n = 1184). There was a negligible and statistically insignificant Spearman's Rho correlation of 0.009 (n = 673, *p* = 0.823) between supermarket access and the LPI.

**Table 6.** Lived Poverty and supermarket access in previous year in Mexico City.

| Lived Poverty Status | no access n (%) | access n (%) |
|---|---|---|
| <=1.00 | 456 (89.6%) | 652 (96.6%) |
| 1.01–2.00 | 47 (9.2%) | 22 (3.3%) |
| 2.01–3.00 | 6 (1.2%) | 1 (0.1%) |
| Total | 509 (100%) | 675 (100%) |

Regular supermarket access did not appear to have a significantly different relationship with the LPI in Nairobi. The variables in Table 7 share a nonrandomly distributed relationship according to a Fisher's exact test of independence at an alpha of 0.05 (F = 50.427, *p* < 0.001, n = 1349). Almost all (92.3%) households with regular access to a supermarket had a score of 1.00 or less on the LPI, compared with 78.4% of those with irregular access.

**Table 7.** Lived poverty and regularity of supermarket access in Nairobi.

| Lived Poverty Status | regular access n (%) | irregular access n (%) |
|---|---|---|
| <=1.00 | 891 (92.3%) | 301 (78.4%) |
| 1.01–2.00 | 71 (7.4%) | 74 (19.3%) |
| 2.01–3.00 | 3 (0.3%) | 8 (2.1%) |
| 3.01+ | 0 (0.0%) | 1 (0.3%) |
| Total | 965 (100%) | 384 (100%) |

Similarly, the regularity of supermarket access in Mexico City did not have a significantly different relationship with the LPI. These variables share a similar nonrandomly distributed relationship according to a Fisher's exact test of independence at an alpha of 0.05 (F = 14.53, *p* < 0.001, n = 1182) (Table 8). 95.4% of households with regular access to a supermarket had an LPI of 1.00 or less, compared with 89.5% with irregular access.

**Table 8.** Lived Poverty and regularity of supermarket access in Mexico City.

| Lived Poverty Status | regular access n (%) | irregular access n (%) |
|---|---|---|
| <=1.00 | 755 (95.4%) | 350 (89.5%) |
| 1.01–2.00 | 33 (4.2%) | 37 (9.5%) |
| 2.01–3.00 | 3 (0.4%) | 4 (1.0%) |
| Total | 791 (100%) | 391 (100%) |

Differences also emerged when examining the relationship between household poverty and supermarket access. In Nairobi, there is a consistently weak but statistically significant correlation between accessing supermarkets more regularly with household levels of poverty, indicating that access to supermarkets may be a good indicator of better lived poverty. In Mexico City, the relationship is inconclusive.

*3.2. Objective Two Results: Test Emerging Concepts of Food Deserts (Food Deserts Plus)*

### 3.2.1. Is There a Relationship between Household Access to All Food Retail Sources and Household Food Security?

This question assumes that household food security status can vary according to the type of household food sources accessed. Table 9 demonstrates that households in Nairobi accessing street sellers and venders had a higher average HFIAS score than those households that accessed supermarkets, fast food outlets, online market shopping, or restaurants. It is important to note that this is a multiple response question.

**Table 9.** Average HFIAS scores by household food sources in previous year in Nairobi.

| Food Sources | n | Mean HFIAS |
|---|---|---|
| Informal street sellers/vendors | 631 | 6.46 |
| kiosk / corner store | 961 | 5.91 |
| Other shops including grocer or butcher | 1144 | 5.83 |
| City Council/County market | 715 | 5.48 |
| Supermarket | 1096 | 4.98 |
| Restaurant | 306 | 3.44 |
| Online market shopping | 12 | 2.83 |
| Fast food outlets | 199 | 1.98 |

Similarly, there were differences in average HFIAS scores across the food sources accessed by the households in Mexico City (Table 10). The highest HFIAS scores were observed among households that accessed food from restaurants or fast food outlets and shopped at convenience stores (4.95) and markets (3.34). Those that accessed food from supermarkets had a lower mean HFIAS score than either (2.32).

**Table 10.** Average HFIAS scores by household food sources in previous year in Mexico City.

| Food Sources | n | Mean HFIAS |
|---|---|---|
| Convenience stores | 43 | 4.95 |
| Market | 1031 | 3.34 |
| Small shop | 816 | 3.1 |
| Street seller/vendor | 195 | 2.82 |
| Supermarket | 681 | 2.32 |
| Online market shopping | 12 | 1.92 |
| Restaurant | 82 | 0.83 |
| Fast food outlets | 54 | 0.26 |

### 3.2.2. Is There a Link between the Type of Food Products Purchased and the Sources of Those Food Products at the Household Level?

One potential reason underlying the distribution of HFIAS scores by food source may have to do with the types of food accessed at these food sources. Supermarkets were the most common place to buy many food items in Nairobi, followed by kiosks, small shops, and street sellers. Items most commonly purchased at supermarkets included maize meal, brown bread, rice, pasta, tinned food, frozen meat, sour milk, tea/ coffee, sugar, cooking oil, snacks, and sweets. Fresh foods, on the other hand, are not often purchased at supermarkets. Items such as fruit and vegetables were commonly purchased at small shops, kiosks, and street traders. Fresh fish, cooked fish, and pies/samosas were most often purchased from street sellers. Fresh and whole foods are therefore most often purchased at smaller scale retail types while more processed foods and foods high in sugar and fat are most often purchased at supermarkets.

In Mexico City, many of the food items recorded in the survey instrument were bought from supermarkets and seem to be supplemented by markets and small shops. One exception was eggs,

with 64% buying them at some point from small shops, 30% from markets, and 26% from supermarkets. Another exception is tamales, quesadillas, and tacos which were purchased primarily from street sellers and then markets. Fresh fish and chicken were primarily purchased at markets, whereas frozen fish and chicken were primarily purchased at supermarkets. Fresh fruit and fresh cooked vegetables were purchased more often from markets than supermarkets. Mexican staples, such as tortillas, were bought from specialized stores, whereas rice was bought more or less equally from markets and supermarkets. Bread was mostly bought in supermarkets, with only a small percentage in markets. Finally, the majority of cooking oil was purchased in supermarkets.

### 3.2.3. Is There a Link between Fruit and Vegetable Purchase/Consumption and Household Food Security?

In Nairobi, there seems to be a statistically insignificant relationship between fruit and vegetable consumption in the previous 24 hours and food security (Table 11). These variables do not share a nonrandomly distributed relationship according to a chi-squared test of independence at an alpha of 0.05 ($x2 = 6.504$, $p = 0.09$, n = 1402).

**Table 11.** HFIAP scores by household fruit and vegetable consumption in Nairobi.

| Food Security Status | None consumed n (%) | Fruit/Veg. consumed n (%) |
|---|---|---|
| Food Secure | 33 (21.6%) | 377 (30.2%) |
| Mildly Food Insecure | 17 (11.1%) | 159 (12.7%) |
| Moderately Food Insecure | 56 (36.6%) | 407 (32.6%) |
| Severely Food Insecure | 47 (30.7%) | 306 (24.5%) |
| Total | 153 (100%) | 1249 (100%) |

Similarly, there seems to be a relationship between fruit and vegetable consumption in the last 24 hours and food security in Mexico City (Table 12). The variables shared a nonrandomly distributed relationship with household food security status according to a chi-squared test of independence at an alpha of 0.05 ($x2 = 20.740$, $p < 0.001$, n = 1201). Again, this relationship is not as strong as it is between food security and supermarket access.

**Table 12.** HFIAP scores by household fruit and vegetable consumption in Mexico City.

| Food Security Status | None consumed n (%) | Fruit/Veg. consumed n (%) |
|---|---|---|
| Food Secure | 75 (36.2%) | 520 (52.3%) |
| Mildly Food Insecure | 25 (12.1%) | 121 (12.2%) |
| Moderately Food Insecure | 31 (15.0%) | 107 (10.8%) |
| Severely Food Insecure | 76 (36.7%) | 246 (24.7%) |
| Total | 207 (100%) | 994 (100%) |

### 3.2.4. Is the Number of Household Food Retail Sources Related to Household Food Security?

In Nairobi, there was a negligible but statistically significant Spearman's Rho correlation of −0.140 (n = 1401, $p < 0.001$) between the number of retail food sources accessed by the household in the last year and household food security status. The sign on this correlation suggests that a higher number of food retail sources is associated with greater household food security, although the correlation effect size is minimal.

In Mexico City, there was also a negligible but statistically significant Spearman's Rho correlation of −0.127 (n = 1200, $p < 0.001$) between the number of food retail sources accessed in the previous year and household food security status. As in the Nairobi survey, it appears it was common among households in Mexico City to have multiple food retail sources.

To summarize, there does appear to be a relationship between household access to food retail types and household food security in both cities. In Nairobi, households accessing street sellers and vendors are more likely to be food insecure than those accessing fast food outlets, online market shopping, or restaurants. Households accessing supermarkets are moderately more food secure than those accessing street vendors. In Mexico City, the strongest relationship is between high levels of food security and visiting restaurants or fast food outlets. Whereas in Nairobi, the most food insecure households accessed food through street sellers and markets; Mexico City households had more variety in food sources.

In both Nairobi and Mexico City, there seems to be a link between the types of food products purchased and the sources of those food products. In Nairobi, supermarkets appear to be the most common place to buy many food items. Fresh or cooked vegetables, however, are most often purchased from markets whereas fresh meats are purchased at butcheries. In Mexico City, supermarkets are the most common place to buy the most items, followed by formal and informal markets. Fresh fruit and fresh cooked vegetables, however, are bought mostly from markets, while fresh meat is bought from these sources or butchers.

In Mexico City, there seems to be a relationship between fruit and vegetable consumption and food security. This relationship is not as strong as it is between food security and supermarket access. Lastly, there was a negligible relationship between the number of food retail sources accessed by households and household food security status in Nairobi, and no statistically significant relationship in Mexico City.

### 3.3. Objective Three Results: Test Crush and Battersby's [3] Definition of Food Deserts (Food Deserts in the Global South)

Is There a Relationship between Access to All Food Sources (Market and Otherwise) and Household Income, Household Dietary Diversity, Food Access/Food Price Challenges?

Table 13 indicates that there was a positive statistically significant relationship between the number of food sources a household accesses and both household income and household dietary diversity in Nairobi. A higher number of food sources was related to improved household food security, dietary diversity, reduced food price impact, and higher household income. While these correlations were statistically significant, their effect sizes were small, indicating a weak relationship between the number of household food sources and each variable. The strongest relationship was observed between household dietary diversity and the number of food sources accessed in the last year, although this relationship is weak (Rho = 0.209).

**Table 13.** Spearman's Rho correlation of HFIAP, Household Dietary Diversity Score (HDDS), household food price impact, and household income with the number of household food sources in the previous year in Nairobi.

| | Number of Food Sources | | |
| --- | --- | --- | --- |
| | **Rho** | **P-Value** | **n** |
| HFIAP | −0.096 ** | <0.001 | 1401 |
| HDDS | 0.209 ** | <0.001 | 1413 |
| Food Price Impact | −0.093 ** | <0.001 | 1396 |
| Household Income Quintiles | 0.186 ** | <0.001 | 830 |

* *p*-value < 0.05, ** *p*-value < 0.01.

A similar set of correlations is observed in Mexico City. Table 14 demonstrated a weak to negligible relationship between household dietary diversity and number of food sources. A higher number of food sources was related to improved household food security, dietary diversity, reduced food price impact and higher household income. However, the correlation effect sizes were small, indicating a

weak correlation relationship. The strongest relationship observed was between household income and the number of household food sources (Rho = 0.305), although this relationship is still weak.

**Table 14.** Spearman's Rho correlation of HFIAP, HDDS, household food price impact, and household income with the number of household food sources in the previous year in Mexico City.

|  | Number of Food Sources | | |
|---|---|---|---|
|  | **Rho** | **P-Value** | **n** |
| HFIAP | −0.113 ** | <0.001 | 1200 |
| HDDS | 0.282 ** | <0.001 | 1209 |
| Food Price Impact | −0.137 ** | <0.001 | 1204 |
| Household Income Quintiles | 0.305 ** | <0.001 | 825 |

\* *p*-value < 0.05, \*\* *p*-value < 0.01.

While there were statistically significant correlations observed in both cities between the total number of household food sources accessed in the last year and household food security, dietary diversity, reduced food price impact, and higher household income, these correlation coefficients tend to be weak. In comparison to testing the relationship between household food retail sources and food security (Section 2.4), adding nonmarket food sources does not seem to have a significant impact on the outcome.

## 4. Conclusions

The food desert concept has proven to be a useful way to raise debate about the structural inequalities in urban food systems overall. However, this paper has demonstrated that applying various conceptions of food deserts to cities in the Global South is potentially problematic. We show that, while these associations are, with some exceptions, statistically significant and show the expected correlation direction between household food sourcing behavior and food security, the strength of these relationships tend to be weak. When assessing the relative utility of the three food desert concepts in the contexts of Nairobi and Mexico City, they appear to be equally inapplicable.

Our findings show that food deserts in the Global South should not be understood through the proxy measurement of supermarket access. Supermarket intervention responses to neighborhood scale issues of food and nutrition insecurity would not reflect the plurality of food sourcing options residents frequent. Rather, households in both Nairobi and Mexico City access large numbers of food retail sources, including informal neighborhood retailers, kiosks, corner stores, small shops, and markets. Policy responses to food insecurity challenges should be culturally and contextually relevant, which necessitates engagement with vital food sourcing options for urban residents other than just modernized retailers such as supermarkets. There remains a need for more debate regarding neighborhood food systems in cities in the Global South, and alternative food desert conceptions might offer a useful way to raise debate about the structural inequalities at this scale. Our findings indicate, however, that the urban food desert concepts tested here may have limited relevance to explaining urban food insecurity in two distinct cities in the Global South.

There are important limitations that accompany the findings from this investigation. First, this investigation should not be interpreted as an analysis of any causal relationships between food sourcing and food security. The methods test the predictive relationship between food source access and food security assumed by the three urban food desert definitions. Therefore, the paper assesses whether food insecurity can be inferred based on limited household access to specific food sources. Additional research will be needed to assess any causal interpretations of urban food deserts. Last, investigating food deserts through an understanding of their complexity requires the inclusion of factors such as mobility, transportation, time, education, structural inequalities, and neighborhood policy environments, which have not been explored in this study.

Given that research on food deserts in the Global South has not yet systematically explored the structural drivers of food insecurity that operate outside the home, future studies should expand to neighborhood and citywide scales. This paper and future studies in the field are relevant to Sustainable Development Goal (SDG) 2—to end hunger—as well as Goal 11 to make cities and human settlements inclusive, safe, resilient, and sustainable. Urban food insecurity dynamics in the South are changing and becoming increasingly problematic. There is little chance of reversing this growth without the development and implementation of sound, evidence based, neighborhood and citywide food security strategies that contribute to the achievement of the SDGs. Future research should systematically explore their dynamics to inform policy addressing food insecurity in cities of the South.

**Author Contributions:** Conceptualization: J.W. and L.H.; Methodology: J.W., L.H., C.M., S.O., G.C. and S.G.A.; Validation: C.M.; Formal Analysis: C.M.; Investigation: J.W. and L.H.; Data Curation: C.M.; Writing—Original Draft Preparation: J.W., L.H. and C.M.; Writing—Review & Editing: J.W., L.H. and C.M.; Supervision: C.M.; Project Administration: J.W., L.H., C.M., S.O., G.C. and S.G.A. Data Collection: S.O., G.C., S.G.A., and C.M.

**Funding:** This research was funded by the Hungry Cities Partnership project supported by the Social Sciences and Humanities Research Council of Canada and the International Development Research Centre through the International Partnerships for Sustainable Societies (IPaSS) Program.

**Acknowledgments:** We wish to acknowledge the support of the following. The International Development Research Centre (IDRC) and the Social Sciences and Humanities Research Council of Canada (SSHRC) for the Hungry Cities Partnership. We wish to thank our colleagues in AFSUN for their assistance and wish to wish to thank the following for their assistance with research planning and implementation: Samuel Owuor, Guenola Capron, Salomón Gonzalez Arellano, Jonathan Crush, Bruce Frayne, Maria Salomone, Gareth Haysom, and Mary Caesar.

**Conflicts of Interest:** The authors declare no conflict of interest.

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
