# Peer review of "Do Urban Food Deserts Exist in the Global South? An Analysis of Nairobi and Mexico City"

_sustainability, doi:10.3390/su11071963_

Round 1
Reviewer 1 Report
This paper provides an overview of research done in two cities of the Global South about the prevalence and effects of food deserts. As the authors note, this is a region where comparatively less work has been done, though there has been more in recent years.
The literature review simplifies a complex and evolving body of literature to something overly linear and straightforward. Just in Geography, there's a mix of folks who still use distance based measures and researchers who provide more nuanced analyses. I don't think it's fair to present it in these easily divided stages. I also don't think the stratification of these research questions along this timeline works as a result. Why test the earliest research questions if the literature review questions the validity of that concept and suggests later research has moved past it? Perhaps say something more along the lines of an ecological model, focusing on multiple elements/scales of factors that contribute to food access? To me, this is the biggest weakness of this article, as the primarily theoretical framework just isn't developed very well.
Also, nothing in the literature review focuses on the regionality of past research, even though that's a main contribution the author(s) cite in doing this research. The lit review should focus more on the geographic context of past research and highlight ways that a focus on the US/UK/etc. might limit research and how work by Battersby and others in the Global South provides a growing supplemental perspective. The specific case studies given
A section after the lit review restating the goals of this study in light of past research would be good. There's some of that in the introduction, already, but more later on would provide better context for the subsequent methods section.
Overall, the conceptual weaknesses of this paper are significant. The three types of "food deserts" here are poorly developed and in some cases inaccurate. I cannot recommend it for further revision or publication as a result.
More specific comments:
p. 1, l. 41: "which has only become...of late" Wording here is awkward. Also the following argument about supermarket expansion cites only one article about South Africa, so making a larger claim about the Global South may be overstepping the evidence.
p. 2, l. 84: "distance to supermarkets" The issue of Urban Studies in 2002 with a lot of the early research actually focused quite a bit on supermarket basket studies and retail geography. There's some use of GIS, but in my read that didn't become more prominent until 2005-2006.
p. 2, l. 90: Need a citation and more detail on "implicating pricing and mobility". This paragraph as a whole is fairly vague--what would evidence look like to confirm variables, for example?
p. 2, l. 98: "add x and stir" is pretty informal for an academic journal article and trivializes what have been significant changes in methodology. The paper also lists pricing as an added factor, but that's also listed in the last section.
p. 3, l. 110: Food oases and food swamps don't really change the spatial conception of food deserts--they just change the ways we describe those spacs. More detail about the exact kind of complexity added would be good--it's pretty general.
p. 4: There needs to be a section here justifying the choice of these study sites and explaining how each provides an opportunity to focus on food access/food deserts.
p. 5, l. 220-222: Not sure why these tests are listed out of context. The later paragraphs explain how/why they're used, which is more helpful.
Top of p. 6: Not sure why these tests are listed out of context. They're later described in stages, but this is still vague: which variables are correlated, for example? All the ones mentioned above? If so, for what purpose? Also the phrase "All shapefiles were removed from the GIS analysis" doesn't make sense to me. And scatterplots are a very basic way to reveal clustering. Getis-Ord or LISA analysis would be a better measure for spatial autocorrelation.
p. 14: The discussion doesn't make this claim explicitly, but there's an implication that access to supermarkets has a causal association with food insecurity--that's how the original concept is being "tested." But a chi-square test can't identify causality.
p. 15: The Crush and Battersby "concept" given is just a single sentence, and the tests referenced in this section don't tackle the complex relationships between people and their food options. There's nothing on mobility or perceptions of neighborhood environment, nor is there anything on policies shaping the food evnironment.
The map figures are missing from my copy of this paper, and so I cannot comment.
Citation 12: Typo in the article title.
Author Response
Thank you for providing a thorough and constructively critical review of the manuscript. We have taken all of your comments seriously, and addressed the majority of them.
"The literature review simplifies a complex and evolving body of literature to something overly linear and straightforward. Just in Geography, there's a mix of folks who still use distance based measures and researchers who provide more nuanced analyses. I don't think it's fair to present it in these easily divided stages. I also don't think the stratification of these research questions along this timeline works as a result. Why test the earliest research questions if the literature review questions the validity of that concept and suggests later research has moved past it? Perhaps say something more along the lines of an ecological model, focusing on multiple elements/scales of factors that contribute to food access? To me, this is the biggest weakness of this article, as the primarily theoretical framework just isn't developed very well."
Thank you for these comments. As you rightly point out, our genealogical method required us to account for a body of literature that has not evolved linearly. For this reason, we have reframed the three definitions of food deserts as to avoid assuming a linear historical evolution.
"p. 1, l. 41: "which has only become...of late" Wording here is awkward. Also the following argument about supermarket expansion cites only one article about South Africa, so making a larger claim about the Global South may be overstepping the evidence."
We have provided further reference to literature arguing that this expansion process is, in fact, global.
"p. 4: There needs to be a section here justifying the choice of these study sites and explaining how each provides an opportunity to focus on food access/food deserts."
Thank you for this suggestion. We have now added a justification for the choice of these case studies.
"Top of p. 6: Not sure why these tests are listed out of context. They're later described in stages, but this is still vague: which variables are correlated, for example? All the ones mentioned above? If so, for what purpose? Also the phrase "All shapefiles were removed from the GIS analysis" doesn't make sense to me. And scatterplots are a very basic way to reveal clustering. Getis-Ord or LISA analysis would be a better measure for spatial autocorrelation."
Thank you for raising this point. The section at the top of page 6 (in the original manuscript) describes the variables that will be analyzed in this manuscript. Many of the variables had to be calculated from raw data in order to accurately represent the phenomena we are investigating. We have also described the purpose of this analysis as a means of testing each definition of urban food deserts.
We take your point about the removal of shapefiles from the GIS analysis and have revised this statement to make it easier to interpret.
Thank you for raising the point about matching the variables with each respective analysis in this section. We have added this description to the manuscript. We also take your point about the GIS analysis. In practice, urban food deserts have been used to map food insecure household locations in cities (for better or worse). The point of this analysis was to demonstrate whether there are sections of cities where households both experience food insecurity/poverty and limited access to supermarkets/retail food sources/all food sources. These GIS analyses are also accompanied by tests of association (Pearson’s Chi-Square and Spearman’s Rho) to assess any statistical association between these variables.
"p. 14: The discussion doesn't make this claim explicitly, but there's an implication that access to supermarkets has a causal association with food insecurity--that's how the original concept is being "tested." But a chi-square test can't identify causality."
Urban food deserts have been used as a tool for predicting where food insecure households may be in a city based on supermarket availability. We are testing whether it is possible to predict food insecurity based on supermarket access. We are not testing whether any causal relationship exist between these variables. We have added a paragraph on limitations to explain this fact.
"p. 15: The Crush and Battersby "concept" given is just a single sentence, and the tests referenced in this section don't tackle the complex relationships between people and their food options. There's nothing on mobility or perceptions of neighborhood environment, nor is there anything on policies shaping the food environment."
Thank you for pointing this out. Investigating food deserts through an understanding of complexity requires the inclusion of factors that have not been accounted for in this study (mobility, transportation, time, education, structural inequalities, and policy environments, which have not been explored in this study. This limitation has now been accounted for within the text.
Reviewer 2 Report
This is a well designed study and drafted manuscript.
Author Response
Thank you for taking the time to review the manuscript.
Reviewer 3 Report
The manuscript is very well- written and approaches an important concept towards meeting the sustainable development golas for erradicating poverty and hunger. Moreover, the authors have compared two different cities of the Global South cities, Nairobi and Mexico City. The manuscript is well presented, and conceptualise with important outcomes of the food deserts concept. I would recommend the acceptance of this manuscript following some minor changes.
Minor comments:
Introduction- L65-L72-This paragraph could be moved to the end of the Introduction, as would be more adequate to outline the objective of the paper. Also, authors could pinpoint the importance of using two cities from different geographical areas, namely Africa and South America.
Material and Methods- In the sub-section Research Objectives and questions, I would advice the authors to include a short paragraphs describing the surveys, and not only presenting the table.
In Analysis sub-section, the software used should be included.
Discussion- I would advise to eliminate the conclusions section, and would merge into the discussion. Also, it would be important to include a paragraph mentioning the applicability of food deserts concept in alignment of SDGs.
Figures 1 and 2 should be in a high-quality format to improve readability.
Author Response
Thank you for taking the time to review this manuscript and provide criticisms. We have taken your review seriously, and have addressed many of your comments.
"Introduction- L65-L72-This paragraph could be moved to the end of the Introduction, as would be more adequate to outline the objective of the paper. Also, authors could pinpoint the importance of using two cities from different geographical areas, namely Africa and South America."
Thank you for suggesting that we discuss the importance of using two cities from two geographical areas and thus distinct contexts. We have added this discussion to the text. The introduction has also been edited as per your recommendation.
"Material and Methods- In the sub-section Research Objectives and questions, I would advice the authors to include a short paragraphs describing the surveys, and not only presenting the table."
We agree with this suggestion and have included a small paragraph describing the surveys.
"In Analysis sub-section, the software used should be included."
We have now included the software used in the analysis sub-section.
"Discussion- I would advise to eliminate the conclusions section, and would merge into the discussion. Also, it would be important to include a paragraph mentioning the applicability of food deserts concept in alignment of SDGs."
Thank you for this comment. We have actually re-organized the manuscript so that the discussion occurs at the end of each objective results sub-section, instead of them all being discussed at the end. We have also included a brief comment noting the relevance of this research to the achievement of the SDG, particularly SDG 2 and 11.
"Figures 1 and 2 should be in a high-quality format to improve readability"
We have provided higher resolution figures to address this issue.
Round 2
Reviewer 1 Report
The authors have made significant revisions to this paper which address some of the concerns raised in my first review. However, the primary two issues identified earlier are still salient for this version of the manuscript.
First, the paper's overall contribution to the broader literature on food access and food security remains unclear. Two main goals are identified--testing three different framings of food deserts and examining the relevance of these frameworks in the global south. Yet the analyses proposed by the authors' own admissions cannot speak to broader household issues shaping food access, which fit within the "food deserts as complexity" framework. Furthermore, these three framings are not used elsewhere in the research to my knowledge, and so the value of testing them remains unclear. On the issue of regionality, while Mexico City and Nairobi seem like promising study sites, the authors provide very limited explanation of the reasons they were chosen other than that they both exist in the global south--an area of several billion people. A more nuanced analyses of how the differences between these two cities would be useful is needed.
Second, while the authors do take greater care in the statistical claims made through these analyses (though some might still question whether "predictive" still implies a kind of causality), the methods used here show only general associations between household shopping and food security. More attention to potential explanatory frameworks that might explain these associations might be useful--in the conclusion if not elsewhere.
One other note not made in the first round of revision--the GIS analysis done here is very much descriptive, visualizing the data and noting the lack of a striking spatial pattern. There is a whole series of techniques in point pattern analysis that might provide a more robust analysis.
Overall, the core issues remaining this paper mean I cannot recommend publication in the articles' current form.
Author Response
We thank the reviewer for the thorough review of the manuscript. We have made revisions based on their comments and criticisms.
With regards to the third research objective, we recognize that we do not test the complexity of food deserts as it was described in the introduction. We have edited the text and sub headers to make it clearer that we are testing Crush and Battersby’s definition of food deserts, which was adapted with the intention of the concept better suiting the food systems of cities in the Global South.
We make no claim that the three food desert labels within this paper (the original food desert, food deserts plus, and food deserts in the global south) are used elsewhere in the literature. They have been developed by the authors based off of framing and measurement approaches used within previous discourse focusing on cities in the UK and North America. There is clear value in testing these concepts in Mexico City and Nairobi because the food systems in these contexts are distinct from cities in the Global North, using both traditional/ informal and highly sophisticated/ modern food procurement systems to meet the food security needs of urban dwellers.
On the issue of regionality, we have added more detail regarding how the differences between these two cities is useful for comparative analysis. We have further elaborated on their unique contexts and development pathways as to further demonstrate the usefulness of our location selection.
With regards to the claim that we are testing causality, this is incorrect. Predictive relationships and causal relationships should not be confused with each other. We have taken great care not to infer any causal claims within our analysis. We have added a paragraph within the conclusion to further elaborate on the implications of our findings for policy and research moving forward.
Last, we recognize that point pattern analysis is a large field with an array of techniques that can be utilised. Without any clear criticism of the techniques this paper employs or a more concise recommendation for an alternative technique, we cannot seriously consider this comment.
Round 3
Reviewer 1 Report
The authors have attempted to revise this paper to better frame their choice of cities and the theoretical constructs they are testing. To my read, these aspects of the paper remain underdeveloped. The link to Crush and Battersby's work is covered in just a single sentence, and while both Nairobi and Mexico City are defined in more detail, the authors only assert that the two cities have "significant differences" without explaining what those are (political/policy differences, agricultural practices, etc.) in any detail. Fleshing each of these out in more detail would provide better insight into how differences across these cities might illustrate the dynamics in cities playing out across the region.
Regarding the mapping and point point pattern analysis, with the large number of points mapped, it is difficult to say for certain what spatial patterns are evidence. The use location quotient or kernel density techniques may help clarify these patterns. Some form of aggregation to area level statistics would allow for more meaningful discussion. Otherwise, I'm not sure the inclusion of these maps is justified.
Author Response
Based on the reviewer's latest report, we have made the following revisions to the manuscript:
1) All spatial analysis figures and associated text has been removed;
2) The link to Crush and Battersby's work has been discussed in more detail, including an explanation as to why food deserts in cities in the Global South might need to be understood differently;
3) The literature review/ justification for case study selection has been expanded further to discuss similarities and differences in roles of the formal and informal food economies in each city as well as the policies and regulations governing these urban food systems.